# A Vacuum Ultraviolet Ion Source (VUV-IS) for Iodide-Chemical Ionization Mass Spectrometry: A Substitute for Radioactive Ion Sources

Yi Ji[1], L. Gregory Huey[1], David J. Tanner[1], Young Ro Lee[1], Patrick R. Veres[2], J. Andrew Neuman[2,3], Yuhang Wang[1], Xinming Wang[4]

[1]School of Earth and Atmospheric Sciences, Georgia Institute of Technology, Atlanta, GA 30332, USA
[2]NOAA Chemical Science Laboratory, Boulder, Colorado, USA, 80305
[3]Cooperative Institute for Research in Environmental Sciences, University of Colorado Boulder, Boulder, Colorado, USA, 80309
[4]State Key Laboratory of Organic Geochemistry, Guangzhou Institute of Geochemistry, Chinese Academy of Sciences, Guangzhou 510640, China

*Correspondence to*: L. Gregory Huey (greg.huey@eas.gatech.edu)

**Abstract.** A new ion source (IS) utilizing vacuum ultraviolet (VUV) light is developed and characterized for use with iodide-chemical ionization mass spectrometers (I⁻-CIMS). The VUV-IS utilizes a compact krypton lamp that emits light at two wavelengths corresponding to energies of ~10.030 and 10.641 eV. The VUV light photoionizes either methyl iodide (ionization potential, IP = 9.54 ± 0.02 eV) or benzene (IP = 9.24378 ± 0.00007 eV) to form cations and photoelectrons. The electrons react with methyl iodide to form I⁻ which serves as the reagent ion for the CIMS. The VUV-IS is characterized by measuring the sensitivity of a quadrupole CIMS (Q-CIMS) to formic acid, molecular chlorine, and nitryl chloride under a variety of flow and pressure conditions. The sensitivity of the Q-CIMS, with the VUV-IS, reached up to ~700 Hz pptv⁻¹, with detection limits of less than 1 pptv for a one minute integration period. The reliability of the Q-CIMS with a VUV-IS is demonstrated with data from a month long ground-based field campaign. The VUV-IS is further tested by operation on a high resolution time-of-flight CIMS (TOF-CIMS). Sensitivities greater than 25 Hz pptv⁻¹ were obtained for formic acid and molecular chlorine, which were similar to that obtained with a radioactive source. In addition, the mass spectra from sampling ambient air was cleaner with the VUV-IS on the TOF-CIMS compared to measurements using a radioactive source. These results demonstrate that the VUV lamp is a viable substitute for radioactive ion sources on I⁻-CIMS systems for most applications. In addition, initial tests demonstrate that the VUV-IS can be extended to other reagent ions by the use of VUV absorbers with low IPs to serve as a source of photoelectrons for high IP electron attachers, such as $SF_6^-$.

## 1. Introduction

Chemical ionization mass spectrometry (CIMS) has been widely used as a powerful tool to measure various atmospheric compounds with high sensitivity and fast time response. CIMS measurements are based on selective ionization of compounds

in air by reagent ions via ion molecule reactions. CIMS using the iodide anion ($I^-$) and its water clusters as reagent ions ($I^-$-CIMS) has been widely used in the measurements of many atmospheric trace gases, e.g. organic and inorganic acids (hydrogen chloride HCl, nitric acid $HNO_3$, formic acid HCOOH, etc.), halogens (bromine oxide BrO, nitryl chloride $ClNO_2$, etc.) and

peroxycarboxylic nitric anhydrides (PANs) (Slusher et al., 2004; Huey, 2007; Phillips et al., 2013; Lee et al., 2014; Liao et al., 2014; Neuman et al., 2016; Liu et al., 2017; Priestley et al., 2018; Bertram et al., 2011; Thornton et al., 2010; Osthoff et al., 2008).

Typically, $I^-$-CIMS systems use a radioactive isotope, usually $^{210}Po$, as an ion source. $^{210}Po$ emits α particles (with an energy

of ~5 MeV) that directly ionize the carrier gas in the ion source to produce secondary electrons. The secondary electrons are thermalized by collisions and react with methyl iodide ($CH_3I$) to form $I^-$ by dissociative electron attachment. The use of radioactive ion sources with the $I^-$-CIMS system is well-established and has several important advantages. For example, radioactive sources are exceedingly reliable and are easy to use, as they require no external power. Radioactive sources often produce relatively clean mass spectra with few interfering masses. However, radioactive sources have several disadvantages

as well. $^{210}Po$ is toxic and is highly regulated which often makes it difficult to transport, store, and use in remote locations. We have recently developed a lower activity $^{210}Po$ ion source that is subject to fewer regulatory restrictions (Lee et al., 2019). However, there remain applications where the use of any radioactivity is very difficult or prohibited. A more subtle disadvantage is that radioactive ion sources emit continuously, which can lead to the build-up of interfering species. For these reasons, it is desirable to find a non-radioactive alternative to efficiently generate $I^-$ and other reagent ions. Electrical discharges

and x-ray ion sources have been used as ion sources in atmospheric pressure chemical ionization mass spectrometers (AP-CIMS) (Jost et al., 2003; Skalny et al., 2007; Kurten et al., 2011; Wang et al., 2017). However, they have not been commonly employed with $I^-$-CIMS systems perhaps due to limited sensitivity and the generation of interfering ions. Recently Eger et al. (2019) developed a promising ion source using a radio frequency (RF) discharge on an $I^-$-CIMS system, providing another alternative to radioactive sources albeit with lower signal levels. However, the RF source also generated high levels of other

ions such as dicyanoiodate anion $I(CN)_2^-$, which can lead to interference but also provide additional pathways for detecting species such as $SO_2$ and HCl.

In this work, we investigate the use of a small krypton (Kr) lamp as a substitute for a radioactive ion source on an $I^-$-CIMS. Similar lamps have been commonly used in atmospheric pressure photoionization-mass spectrometry (e.g. Kauppila et al.,

2017). The vacuum ultraviolet (VUV) light is generated from two emission lines centered at 116.486 (photon energy = 10.641 eV) and 123.582 nm (photon energy = 10.030 eV). $CH_3I$ has a large absorption cross section ($7\times10^{-17}$ $cm^2$ $molecule^{-1}$) at these wavelengths and a relatively low ionization potential (IP = $9.54 \pm 0.02$ eV) (Holmes and Lossing, 1991; Olney et al., 1998). Absorption of the VUV light by $CH_3I$ forms cations and relatively low energy photoelectrons which can then attach to $CH_3I$ to form $I^-$. Benzene ($C_6H_6$) can also serve as a VUV absorber to produce photoelectrons as it has a larger absorption cross

section ($4\times10^{-17}$ $cm^2$ $molecule^{-1}$) and an even lower IP ($9.24378 \pm 0.00007$ eV) (Nemeth et al., 1993; Capalbo et al., 2016). We

explore the use of $C_6H_6$ as a source of photoelectrons as we have found that delivering even modest quantities of gas phase $CH_3I$ to our ion source can be problematic as it has a tendency to polymerize to non-volatile species in compressed gas cylinders. Tests of $C_6H_6$ as a photoelectron source in this work are performed as it may enhance ion production when used in combination with lower levels of $CH_3I$. $C_6H_6$ may also be used as an electron source for use with other electron attaching compounds with higher IPs, such as $SF_6$, to form reagent ions such as $SF_6^-$.

The performance of the VUV-IS was characterized by measuring the sensitivities on a quadrupole CIMS (Q-CIMS) to formic acid, chlorine ($Cl_2$), and nitryl chloride ($ClNO_2$), under different flow conditions with varying levels of $CH_3I$. Similar tests were also performed using a flow of both $C_6H_6$ and $CH_3I$ through the ion source. Potential interferences due to the VUV light interacting with air or surfaces are investigated by comparing ambient mass spectra obtained with a VUV-IS and a standard 20 mCi $^{210}$Po ion source (NRD Static Control, P-2031) on both a commercial high resolution time-of-flight CIMS (TOF-CIMS) and a Q-CIMS. We also test the potential of measuring PAN using this VUV-IS on a TD (thermal dissociation)-CIMS. The reliability of the VUV-IS is tested by performing field measurements for a six week time period at a remote location. The potential of extending the use of VUV-IS to $SF_6^-$ and airborne operation is also explored.

## 2. Materials and Methods

### 2.1 Quadrupole $I^-$-CIMS and experimental configurations

The Q-CIMS used here is very similar to the system that has measured a variety of species such as $Cl_2$, BrO, and PAN, and has been detailed in previous publications (Slusher et al., 2004; Liao et al., 2011; Liao et al., 2014; Lee et al., 2019). Details specific to these experiments are described below. A diagram of the $I^-$-CIMS system and the experimental layout is shown in Figure 1. Varying levels of calibration standard were added to 4-10 standard liters per minute (slpm) of $N_2$ and delivered to the sampling inlet of the CIMS through perfluoroalkoxy (PFA) Teflon tubing, with dimensions of 1.27 cm outer diameter, and 0.95 cm inner diameter. Approximately 1.6 slpm of this flow was sampled into the CIMS flow tube and the rest was exhausted into the lab. The flow tube was humidified by adding 20 standard cubic centimeters per minute (sccm) of $N_2$ through a water bubbler kept in an ice bath. The flow tube was operated at a pressure of either 20 or 40 Torr by using either a 0.91 mm or 0.635 mm orifice between the flow tube and the collisional dissociation chamber (CDC). The scroll pump flow was controlled to maintain the flow tube at 20 or 40 Torr.

Mass spectra of ambient Atlanta air were obtained, in order to check for potential interferences due to the VUV-IS. For these experiments air was sampled from the roof of the Environmental Science and Technology building on the Georgia Tech campus. A PFA Teflon tube of 0.95 cm inner diameter and 8 m long was used as a sampling inlet. A total flow of ~7 slpm of ambient air was drawn through the inlet, of which 1.7 slpm was sampled into the CIMS flow tube and the rest was exhausted through a diaphragm pump. The flow tube was controlled at 20 Torr.

## 2.2 Calibration sources

Permeation tubes (KIN-TEK Laboratories, Inc.) were used as the sources of $Cl_2$ and formic acid for tests on the Q-CIMS. The output of formic acid tube was measured by ion chromatography (Metrohm Herisau, 761 Compact IC) as detailed by Nah et al. (2018). The $Cl_2$ permeation tube emission rate was measured by conversion to $I_3^-$ in aqueous solution, and the resulting $I_3^-$ was quantified by optical absorption at 352 nm on a spectrophotometer (Finley and Saltzman, 2008). The permeation rates were measured to be $104.7 \pm 7.8$ ng min$^{-1}$ for formic acid and $14.8 \pm 1.2$ ng min$^{-1}$ for $Cl_2$. A $ClNO_2$ standard was generated by passing a humidified flow of $Cl_2$ from the permeation tube in $N_2$ through a bed of sodium nitrite ($NaNO_2$). The yield of $ClNO_2$ from $Cl_2$ was assumed to be 50% as we have consistently found in previous studies (Liu et al., 2017).

Sensitivity tests to formic acid and $Cl_2$ on a TOF-CIMS were performed by standard addition in laboratory air. The calibration species were obtained from a calibrated formic acid permeation device (47 ng min$^{-1}$) and a 4 ppm $Cl_2$ in $N_2$ compressed gas cylinder. The $Cl_2$ cylinder was calibrated by cavity ring-down spectroscopy (CRDS) at 405 nm. The permeation rate of formic acid was measured by catalytic conversion to $CO_2$ followed by $CO_2$ detection as detailed by Veres et al. (2010).

## 2.3 VUV ion source (VUV-IS)

In a typical I$^-$-CIMS system, a flow of $CH_3I$ in $N_2$ passes through a $^{210}Po$ radioactive ion source, to form the reagent ion I$^-$. In this study, the radioactive source was removed and replaced with a VUV lamp assembly. The Kr lamp is powered by a 4 W DC power supply (UltraVolt® AA Series High-Voltage Biasing Supplies). Two configurations (a and b as shown in Figure 1) of the VUV lamp assembly were tested. In both configurations a small Kr VUV lamp (Heraeus, Type No. PKS 106) (19.6 mm diameter × 53.5 mm length) was used to generate ions. This lamp is commonly used in small commercial VOC detectors that utilize photoionization as a detection method. The lifetime of this lamp is estimated to be 4000 hours (~5.5 months of continuous use) by the manufacturer. The VUV lamp was operated at ~280 Volts DC and typically drew ~0.7 mA. In general, the ion current reaching the mass spectrometer increased with increasing lamp voltage (see Figure S1). The VUV lamp was attached to a custom QF 40 centering ring with vacuum epoxy. The centering ring has a thru hole (11.4 mm diameter) with a counterbore (41.1 mm diameter) in the center. The lamp is sealed to the edge of the counterbore with vacuum epoxy. The thru hole allows light from the lamp to enter the ion source region of the CIMS. The QF 40 centering ring is mated on the low pressure side to an inline tee with two QF 40 flanges on the ends and a 0.635 cm NPT fitting in the center. In lamp configuration (a), the inline tee is attached to a standard short QF 40 nipple (126 mm length) which serves as a photoionization region and provides a direct path for the VUV photons and generated ions into the flow tube. In configuration (b), a QF 40 90 degree elbow was attached between the QF 40 nipple and CIMS flow tube to prevent direct exposure of the flow tube to the VUV photons. The ambient pressure side of the QF 40 centering ring, on which the VUV lamp attached, is mated to a QF 40 × QF 16 × QF 40 reducing tee housing for protection of the VUV lamp. The QF 16 branch of the tee enables visual inspection of the lamp operation.

The sensitivity of the VUV-IS for measuring formic acid, $Cl_2$ and $ClNO_2$ was measured for varying $CH_3I$ concentrations and at two flow tube pressures (Table 1 and 2). The impact of adding $C_6H_6$ was investigated by varying the concentration of $C_6H_6$ at a lower level of $CH_3I$. Compressed gas cylinders of ~700 ppmv of $CH_3I$ and ~0.1% of $C_6H_6$ in $N_2$ were used as $CH_3I$ and $C_6H_6$ sources. A variable flow of $N_2$ containing $CH_3I$ was delivered to the ion source to determine the optimal flow (Figure 2). The total ion source flow was regulated at 1 slpm for lamp configuration (a) and 1.2 slpm for lamp configuration (b) in all sensitivity and interference tests. Mixing ratios of $CH_3I$ and $C_6H_6$ mentioned in the following sections are the mixing ratios in the total ion source flow (1 or 1.2 slpm).

### 2.4 TOF-CIMS

The VUV-IS was also characterized by operation on a commercial TOF-CIMS (Aerodyne Research Incorporated) (Lee et al., 2014; Veres et al., 2020). The ion molecule reactor (IMR) used here was constructed from a 150 mm long QF 40 adapter tee with a 9.5 mm fitting in the center, to allow mounting of the VUV-IS. The VUV lamp mounted to the QF 40 inline tee (section 2.1) was attached to a QF 40 × QF 16 conical adapter connected to a flange with a 9.5 mm stainless steel tube. This allowed the VUV-IS assembly to be mated directly to the IMR using standard vacuum components. The IMR was maintained at a pressure of 30 Torr and operated at a total flow of 2.2 slpm. A 1 slpm $N_2$ flow with 30-400 ppmv $CH_3I$ passed through the VUV-IS into the IMR and mixed with 1.2 slpm of ambient air. Water was dynamically added to the IMR to maintain a constant ratio of $I^-$ to $I^-(H_2O)$. This provided real-time compensation for changes in ambient humidity to minimize fluctuations in sensitivity. Mass spectra were obtained with both a VUV-IS and a standard radioactive ion source, sampling ambient air in Boulder Colorado.

### 2.5 TD-CIMS

The sensitivity of TD-CIMS with a VUV-IS (Slusher et al., 2004) was also tested for PAN. The configuration of the TD-CIMS system used in this work is almost identical to that described in Lee et al. (2019) with the radioactive source replaced with the VUV-IS in configuration (b). A known amount of PAN was generated using a photolytic source similar to that described by Warneck and Zerbach (1992). A calibration standard of 1 ppbv of PAN was produced by adding the output of the photolytic source to PAN free ambient air. PAN free air was generated by passing ambient air through a QF 40 nipple filled with stainless steel wool heated to 150 °C (Flocke et al., 2005).

### 2.6 $SF_6^-$-CIMS

The instrument used to test the VUV-IS with $SF_6^-$ as a reagent ion is nearly identical to that used previously to measure BrO on the NCAR GV research aircraft (Chen et al., 2016). The operating parameters of the instrument are very similar to those used previously to simultaneously detect sulfur dioxide, formic, and acetic acid (Nah et al., 2018). However, in this application

the radioactive ion source was replaced with a VUV-IS in configuration (b). The system was periodically calibrated in flight by adding a known amount of isotopically labeled $^{34}SO_2$ into the sampled air flow.

## 3. Results

### 3.1 Q-CIMS Sensitivities using $CH_3I$

The sensitivities and LODs for formic acid, $Cl_2$ and $ClNO_2$ under different experimental conditions using lamp configuration
(b) are compiled in Table 1. With the flow tube at 20 Torr, sensitivities to formic acid, $Cl_2$ and $ClNO_2$ reached up to 147, 161 and 154 Hz pptv$^{-1}$, respectively, using up to 86.5 ppmv of $CH_3I$ in the ion source flow. At 40 Torr, similar sensitivities (128, 149 and 148 Hz pptv$^{-1}$ for formic acid, $Cl_2$ and $ClNO_2$, respectively) were achieved with less $CH_3I$ (19.0 ppmv). Figure 3 shows the dependence of the CIMS sensitivities on the $CH_3I$ level at 20 Torr (upper left) and 40 Torr (lower left) with no other absorber added. In general, CIMS sensitivities increase with the $CH_3I$ mixing ratio. However, the response is less than linear
and appears to saturate at higher levels of absorber. With the maximum concentration of $CH_3I$ (86.5 ppmv, $5.70 \times 10^{13}$ molecule cm$^{-3}$) used in this experiment, ~8% of photons emitted from the VUV lamp were absorbed (see SI for sample calculation). This indicates that other factors such as ion recombination and wall loss limit the ion abundance. The sensitivities and LODs under similar experimental conditions using lamp configuration (a) are show in Table 2 and Figure 4. The sensitivities to formic acid, $Cl_2$, and $ClNO_2$ approached ~700 Hz pptv$^{-1}$, with limits of detection of less than 1 pptv for a one minute integration
period. Limits of detection are defined at a signal-to-noise ratio of 3 where the noise is the variance of the background measurements. In general, the sensitivities using lamp configuration (a) were about a factor of four larger than in configuration (b). Note that all the Q-CIMS sensitivities reported in this work are not normalized to the reagent ion signal, since the reagent ion signal is not known accurately. We estimate the reagent ion signals are ~100 MHz at the highest sensitivities, but the Q-CIMS detector counts ions linearly only up to about 0.5 MHz.

**3.2 Q-CIMS Sensitivities using $CH_3I$ and $C_6H_6$**

In order to test the effectiveness of the addition of another absorber to generate photoelectrons, mixtures of $CH_3I$ and $C_6H_6$ were added to the ion source. Lower mixing ratios (8.8 ppmv at 20 Torr, 1.8 ppmv at 40 Torr) of $CH_3I$ were used in combination with varying amounts of $C_6H_6$ to assess the impact of the addition of $C_6H_6$ to the generated ion current using lamp configuration (b). The sensitivity dependence on $C_6H_6$ mixing ratio is shown in Figure 3. At 20 Torr, up to 229.2 ppmv $C_6H_6$ was added to
8.8 ppmv $CH_3I$ to achieve the equivalent sensitivity (158, 157, and 152 Hz pptv$^{-1}$ for formic acid, $Cl_2$ and $ClNO_2$, respectively) using 86.5 ppmv $CH_3I$ alone. At 40 Torr, up to 58.9 ppmv $C_6H_6$ was added to 1.8 ppmv of $CH_3I$ to reach the maximum level of sensitivities (157, 166, and 138 Hz pptv$^{-1}$ for formic acid, $Cl_2$ and $ClNO_2$, respectively) when using 19.0 ppmv of $CH_3I$. Similar trends for the addition of $C_6H_6$ using lamp configuration (a) was observed and are shown in Table 2 and Figure 4. At both 20 and 40 Torr sensitivities of more than 600 Hz pptv$^{-1}$ were obtained for all species in configuration (a). These results

demonstrate that addition of an absorber (e.g. $C_6H_6$) enables high sensitivity with the VUV-IS while adding typical levels (a few ppmv) of an electron attaching compound.

### 3.3 Q-CIMS Interference Tests

Representative mass spectra (m/z = 20 - 220 amu) taken with a 20 mCi $^{210}Po$ standard radioactive source and the VUV-IS with configuration (a) on an $I^-$-CIMS sampling ambient air are shown in Figure 5. Note that the $I^-$ signal is saturated in all mass spectra due to the very high signal levels. Clearly, the VUV-IS in configuration (a) generates many additional ions compared to a radioactive source. Large signals (> 100,000 Hz) are observed at $O_2^-$ (m/z = 32 amu), $NO_3^-$ (m/z = 62 amu), and $CO_3^-$ (m/z = 60 amu). This indicates that photoelectrons generated on the illuminated surface of the flow tube in the presence of the sampled air leads to formation of $O_2^-$ by electron attachment to $O_2$. This also leads to the formation of $CO_3^-$ and $NO_3^-$ by subsequent reactions with $CO_2$, $O_3$, and $NO_2$ (Mohler and Arnold, 1991). Consequently, the generation of $O_2^-$ initiates significant secondary chemistry that may lead to interfering signals at a large number of masses.

Using the VUV-IS in configuration (b) prevents direct illumination of the flow tube by the VUV lamp and produces similar spectra to those obtained with a radioactive source (Figure 5). The $O_2^-$ signal levels are lower by more than three orders of magnitude compared to configuration (a) where the flow tube is directly illuminated.

Finally, Figure 5 also has a mass spectrum using the VUV-IS in configuration (b) with the addition of 110 ppmv of $C_6H_6$ and 8.8 ppmv of $CH_3I$ in the ion source flow. The addition of the $C_6H_6$ does not produce significant amounts of new ions and the mass spectrum is very similar to that obtained without $C_6H_6$. This indicates that using $C_6H_6$ (or other low IP compounds such as toluene or propene) as a light absorber to generate photoelectrons has the potential to extend the use of the VUV-IS to other electron attaching compounds (e.g. $SF_6$, $HNO_3$, etc.) that have small absorption cross sections in the VUV or have ionization potentials higher than 10.6 eV.

### 3.4 Q-CIMS Field Tests

Ground-based measurements of $ClNO_2$, dinitrogen pentoxide ($N_2O_5$) and formic acid using the $I^-$-CIMS with the VUV-IS were conducted at a rural site in Dongying, China, during the Ozone Photochemistry and Export from China Experiment (OPECE) from March 20 to April 22, 2018. The $I^-$-CIMS was deployed in a shelter, with neither heating nor air conditioning, in a remote location in a bird sanctuary in the Yellow River Delta. The site experienced intermittent power interruptions and large ambient temperature variations, from -2.5 to 29.1 °C, with the temperature inside the shelter ranging from ~10 to 40 °C.

The primary goal for the $I^-$-CIMS during the OPECE campaign was to measure halogen containing compounds. However, we found the presence of halogens to be intermittent. Figure 6 shows representative observations of $ClNO_2$ and $N_2O_5$ for this site when halogens were observed. This figure is consistent with the expected behavior of $ClNO_2$ and $N_2O_5$, both accumulate

during night time, and both decrease after sunrise due to photolysis of $ClNO_2$ and thermal decomposition of $N_2O_5$ followed by photolysis of $NO_3$. $ClNO_2$ is a product of reaction between $N_2O_5$ and chloride containing aerosol (R1, Finlayson-Pitts et al., 1989), and $ClNO_2$ and $N_2O_5$ are well correlated ($R^2 = 0.94$, Figure 6(2)) during the night (6 pm, April 12 to 6 am, April 13).

These measurements of $ClNO_2$ and $N_2O_5$ indicate the performance of the $I^-$-CIMS with the VUV-IS is sufficient to capture atmospheric levels and variability.

$$N_2O_5 + Cl^-_{(aq)} \rightarrow ClNO_2 + NO_3^-{}_{(aq)} \tag{R1}$$

Formic acid which is routinely measured by $I^-$-CIMS and ubiquitous in the atmosphere as both an emission and a secondary

chemical product was also monitored during the campaign. These observations demonstrate that $I^-$-CIMS with a VUV-IS could be operated continuously for an extended period (Figure 7). A clogged mass flow controller on the inlet and a scroll pump failure caused brief measurement interruptions on March 27 and in early April. We could not obtain gas mixtures of $CH_3I$ at this field location, so we used a liquid reservoir as $CH_3I$ source. This led to using $CH_3I$ levels of hundreds of ppmv which may have accelerated degradation of the scroll pump tip seals. We also encountered some temperature control issues and power

interruptions during the mission. However, no direct problems were encountered with the VUV-IS. Online calibration of formic acid was performed every 30 minutes during the mission. The CIMS sensitivity to formic acid was measured to be $185.2 \pm 48.3$ Hz pptv$^{-1}$ during the first day, and $180.5 \pm 24.3$ Hz pptv$^{-1}$ a month later, so we did not notice any drop in sensitivity that could be attributed to a decrease in light intensity from the lamp. In addition, we have used the same VUV-IS since the OPECE field mission (Spring 2018) through early 2020 for both lab studies and field measurements, and have not found

obvious sensitivity degradations.

### 3.5 TOF-CIMS Tests

The VUV-IS was found to give very similar signal levels to those obtained with a standard radioactive ion source (with an activity of ~16 mCi of $^{210}Po$) on a TOF-CIMS. Both sources gave total reagent ion signals of ~10 MHz and similar sensitivity to both formic acid and $Cl_2$ of about 20-25 Hz/ppt. The $I^-$ signal for both sources was ~6 MHz and the $I^-(H_2O)$ signal was ~3

245    MHz. In addition, as illustrated in Figure 8, both sources gave very similar mass spectra for ambient air. The largest difference between the mass spectra were that the VUV-IS gave higher levels of $I_3^-$ while the radioactive source gave higher levels of peaks corresponding to nitric acid (i.e. $I^-(HNO_3)$ and $NO_3^-$). Sensitivities as a function of $CH_3I$ mixing ratio in the flow tube were also tested (Figure S2). Sensitivities to formic acid and $Cl_2$ increase with the $CH_3I$ mixing ratio up to ~100 ppmv. The sensitivities normalized to the reagent ion $I^-(H_2O)$ do not change with $CH_3I$ mixing ratio and are the same obtained with the

radioactive source. These results demonstrate that a VUV-IS can be used on a TOF-CIMS to obtain the same sensitivity, selectivity, and ion distribution as with a radioactive ion source.

### 3.6 PAN Measurement Tests

Preliminary tests using the Q-CIMS as a TD-CIMS demonstrated the potential of the VUV-IS for use in the measurement of PAN. Mass spectra of ambient air with and without PAN are shown in Figure S3. The sensitivity towards PAN was observed to be 49.4 Hz/pptv with an LOD (Signal to noise ratio = 3:1) of 0.64 pptv for a 1 minute integration. No significant interferences were observed during the ambient air tests.

### 3.7 SF$_6^-$-Q-CIMS Tests

Preliminary aircraft-based measurements of sulphur dioxide, formic and acetic acid using the SF$_6^-$-CIMS with the VUV-IS were conducted during an Asian Summer Monsoon Chemical and Climate Impact Project (ACCLIP) test flight based out of Broomfield, Colorado on January 30, 2020. Time series of formic and acetic acid signals are shown in Figure S4(1). The signals of formic and acetic acid are correlated ($R^2 = 0.63$, Figure S4(2)) as observed in previous studies (Souza and Carvalho, 2001; Paulot et al., 2011; Nah et al., 2018). We did not perform online calibrations for formic or acetic acid. Sensitivities to formic and acetic acids during this test flight are estimated to be 5-20 Hz/pptv based on online calibrations of $^{34}$SO$_2$ and measurements of the ratio of their sensitivities (Nah et al., 2018).

### 4. Discussion

The VUV-IS can generate I$^-$ ions and mass spectra that are very similar to a radioactive ion source on both a TOF-CIMS and a Q-CIMS. For this reason, we think that the VUV-IS can replace radioactive ion sources in most I$^-$-CIMS applications without any loss of measurement performance. Perhaps, the largest benefit of the VUV-IS is that it will expand the use of I$^-$-CIMS to locations or situations where radioactivity is not allowed. The VUV-IS is also likely to be useful for laboratory flow tube or chamber studies that use CIMS as a detector (D'Ambro et al., 2017; Huang et al., 2017; Faxon et al., 2018). We have not found any interferences or issues that would limit application of a VUV-IS, though its application to measurement of individual species must be confirmed by further testing.

Using the VUV-IS requires attention to the possibility of interferences caused by the generation of photoelectrons from surfaces that can be attached by oxygen or other compounds in the sampled gas matrix. We were able to minimize this effect by using a flow geometry that shielded the flow tube from the VUV photons albeit at the expense of more than a factor of four in signal on the Q-CIMS. Similarly, the implementation of the VUV-IS on the TOF-CIMS successfully limited O$_2^-$ production by passing the ion source flow through a small diameter tube that discriminated against light reaching the IMR. This issue might also be addressed by further improvements in geometry, the use of optical focusing elements, or higher levels of an absorber molecule.

The VUV-IS source requires an absorbing species to serve as a source of photoelectrons. Extending the use of the VUV-IS to other reagent ions such as $SF_6^-$ (Nah et al., 2018), $Br^-$ (Sanchez et al., 2016), $NO_3^-$ (Eisele and Tanner, 1993), $CF_3O^-$ (Crounse et al., 2006), and $CH_3CO_2^-$ (Veres et al., 2008) requires the addition of an absorber such as $C_6H_6$ or $C_7H_8$. Although the use of $CH_3I$ to produce $I^-$ with the VUV-IS does not require an additional absorber, this application requires relatively high levels of $CH_3I$ to obtain maximum sensitivities. Since $CH_3I$ is a hazardous gas, higher levels of $CH_3I$ could be problematic for some situations such as deployment in a highly regulated environment such as an aircraft. Higher levels of $CH_3I$ may also necessitate protection of scroll pumps with scrubbers and traps. For these reasons, it may be preferable to use a low activity (Lee et al., 2019) or standard radioactive ion source for some applications.

One potential advantage of the VUV-IS is demonstrated by the lower background signals at masses corresponding to nitric acid ($HNO_3$). This is probably due to lower rates of nitrogen radical generation in the VUV-IS as the generated photons are much lower in energy (~10 eV) relative to the alpha particles (~5.4 MeV) from a $^{210}Po$ radioactive source. Additionally, the VUV-IS does not produce ions when power is removed but radioactive decay is continuous. Consequently, interfering species can be continuously generated in a radioactive source, leading to build up of condensable species such as nitric acid. For these reasons, the VUV-IS may generate significantly lower levels of interferences than a radioactive source.

## 5. Summary and Conclusions

The sensitivity of the VUV-IS on the Q-CIMS can reach 100s of Hz pptv$^{-1}$ (Table 1 and 2) similar to the best sensitivity obtained with radioactive sources (e.g. Lee et al., 2019). Care must be taken to avoid illuminating surfaces in the CIMS exposed to air. The VUV-IS requires significantly higher levels of $CH_3I$ than used in radioactive sources (typically ~1 ppmv, Slusher et al., 2004) (Figure 3 and 4), because the $CH_3I$ must also serve as a source of photoelectrons. $C_6H_6$ and other species can be used as a VUV absorber to generate photoelectrons (Figure 3 and 4) without generating excessive interferences (Figure 5). Tests on a TOF-CIMS demonstrated that the VUV-IS and a standard radioactive ion source produced the same reagent ion ($I^-$) abundance. In addition, both sources generated similar mass spectra of ambient air, demonstrating that the VUV-IS source did not produce significant interferences to the detection of most trace gases (Figure 8). Preliminary tests also indicate the VUV-IS is compatible with TD-CIMS and $SF_6^-$-CIMS methods as well as airborne operation.

The VUV-IS described in this work provides sensitivities and limits of detections that are at least comparable to those obtained with a radioactive ion source for both Q-CIMS and TOF-CIMS using $I^-$ as a reagent ion. The VUV-IS is reliable and can be safely deployed in remote field missions. These results demonstrate that the VUV-IS can eliminate the need for radioactivity with an $I^-$-CIMS for most applications. The use of low IP absorbers, such as $C_6H_6$, to generate photoelectrons in conjunction with high IP electron attaching compounds allows the generation of other reagent ions such as $SF_6^-$, $CF_3O^-$, and $NO_3^-$. In summary, the VUV-IS has the potential to eliminate most of the use of radioactivity with CIMS instruments.

*Data Availability*

All of the data used in this manuscript is available upon request of the corresponding author.

*Author Contributions*

Yi Ji performed all of the Q-CIMS experiments and wrote the manuscript with assistance from Greg Huey. David Tanner assisted with all the experiments. Xinming Wang organized the OPECE campaign and provided the $j_{NO_2}$ data. All of the authors were involved in data interpretation and commented on the manuscript. The TOF-CIMS experiments were performed

by Patrick Veres, Andy Neuman, and Greg Huey.

*Competing Interests*

The authors declare that they have no conflicts of interest.

*Acknowledgements*

This work was also supported by NSF grant 1853930 and by NASA grant NNX15AT90G. The OPECE field mission was

supported by NSF grant 1743401. This work was also supported in part by an EPA STAR grant R835882 awarded to the Georgia Institute of Technology. It has not been formally reviewed by the EPA. The views expressed in this document are solely those of the authors and do not necessarily reflect those of the EPA. EPA does not endorse any products or commercial services mentioned in this publication and EPA Grant. We thank Jianhui Tang for providing the RH data on the field. We also thank the Yellow River Delta Ecological Research Station of Coastal Wetland, which belongs to Yantai Institute of Coastal

Zone Research, Chinese Academy of Sciences, for logistical support for the OPECE campaign.

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

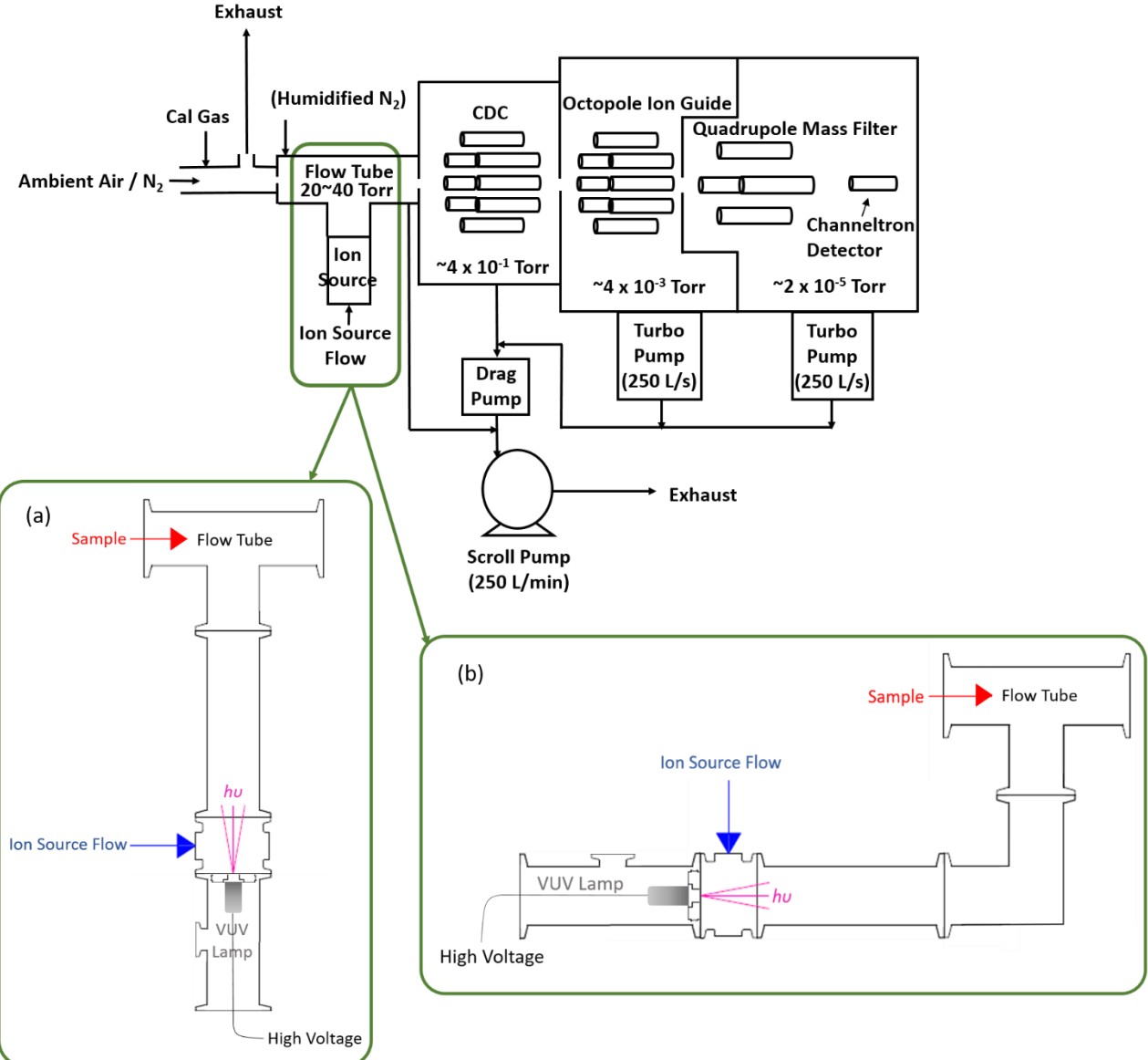

**Figure 1. Diagram of the I⁻-Q-CIMS system with a VUV-IS in two different configurations. Configuration (a) provides the most direct route for the generated ions but also directly illuminates the flow tube. Configuration (b) shields the flow tube from the VUV photons by inserting a QF 40 elbow between the photoionization region and the flow tube.**


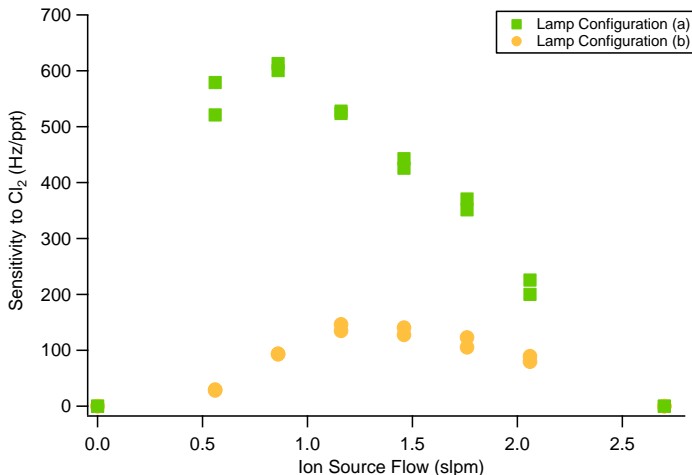

**Figure 2. Q-CIMS sensitivity to $Cl_2 \cdot I^-$ (m/z = 197 amu) as a function of total ion source flow using the VUV-IS with lamp configuration (a) and (b).**

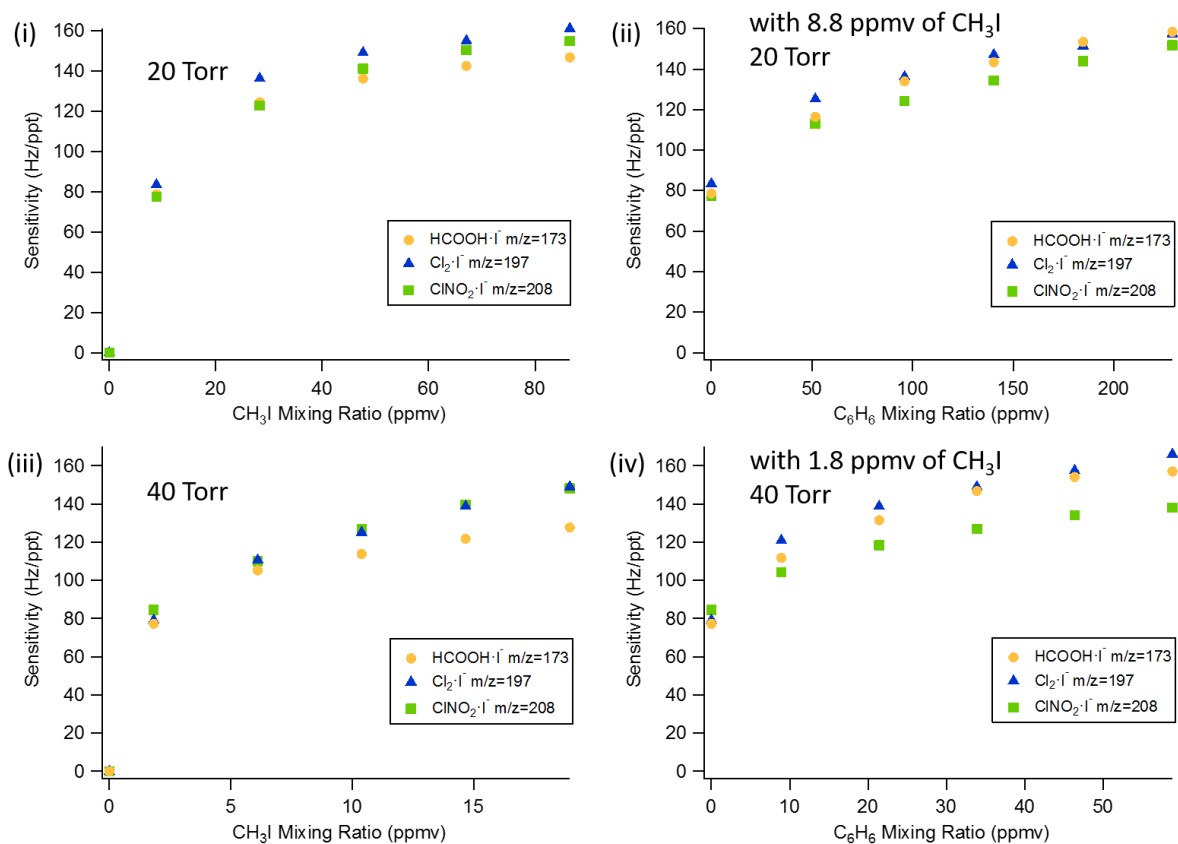

**Figure 3. Q-CIMS using VUV-IS configuration (b): (i) sensitivity as a function of $CH_3I$ at 20 Torr. (ii) sensitivity as a function of $C_6H_6$ at 20 Torr with 8.8 ppmv of $CH_3I$. (iii) sensitivity as a function of $CH_3I$ at 40 Torr. (iv) sensitivity as a function of $C_6H_6$ at 40 Torr with 1.8 ppmv of $CH_3I$.**

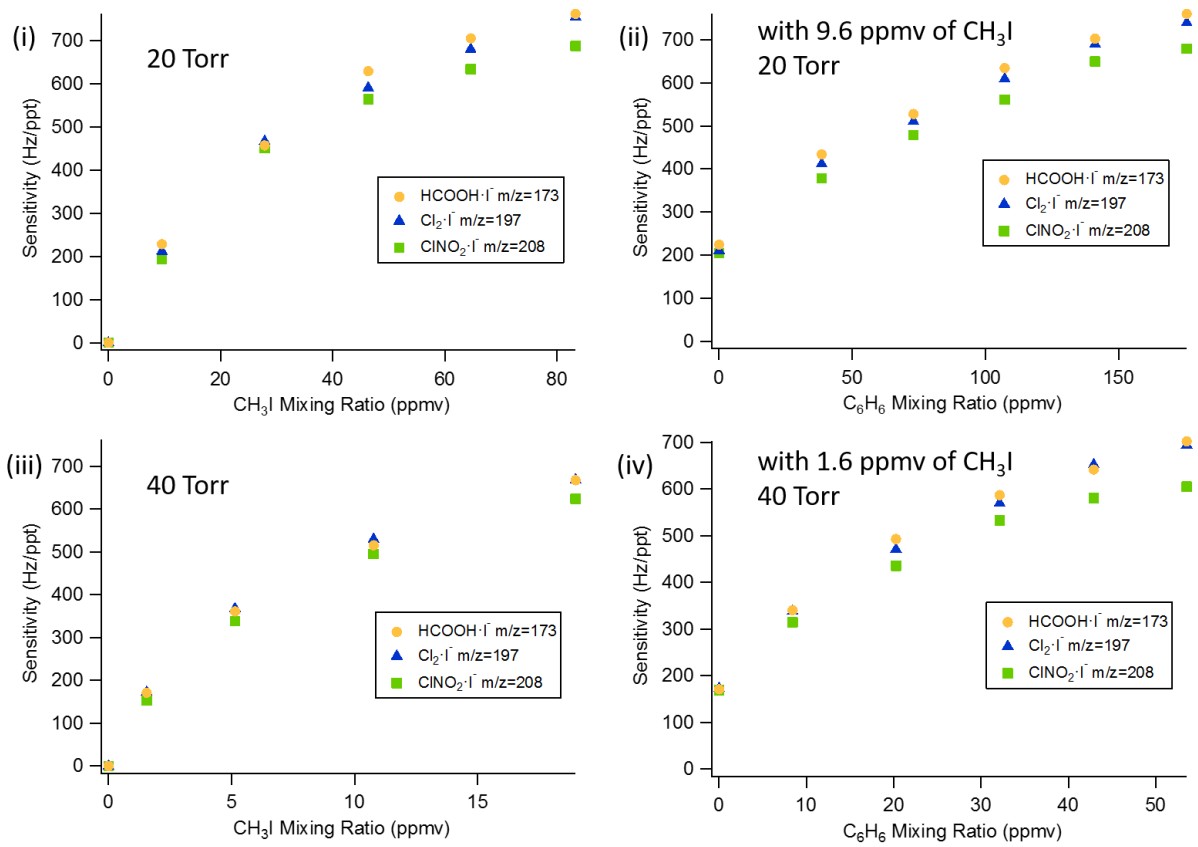

**Figure 4. Q-CIMS using VUV-IS configuration (a): (i) sensitivity as a function of CH₃I at 20 Torr. (ii) sensitivity as a function of C₆H₆ at 20 Torr with 9.6 ppmv of CH₃I. (iii) sensitivity as a function of CH₃I at 40 Torr. (iv) sensitivity as a function of C₆H₆ at 40 Torr with 1.6 ppmv of CH₃I.**


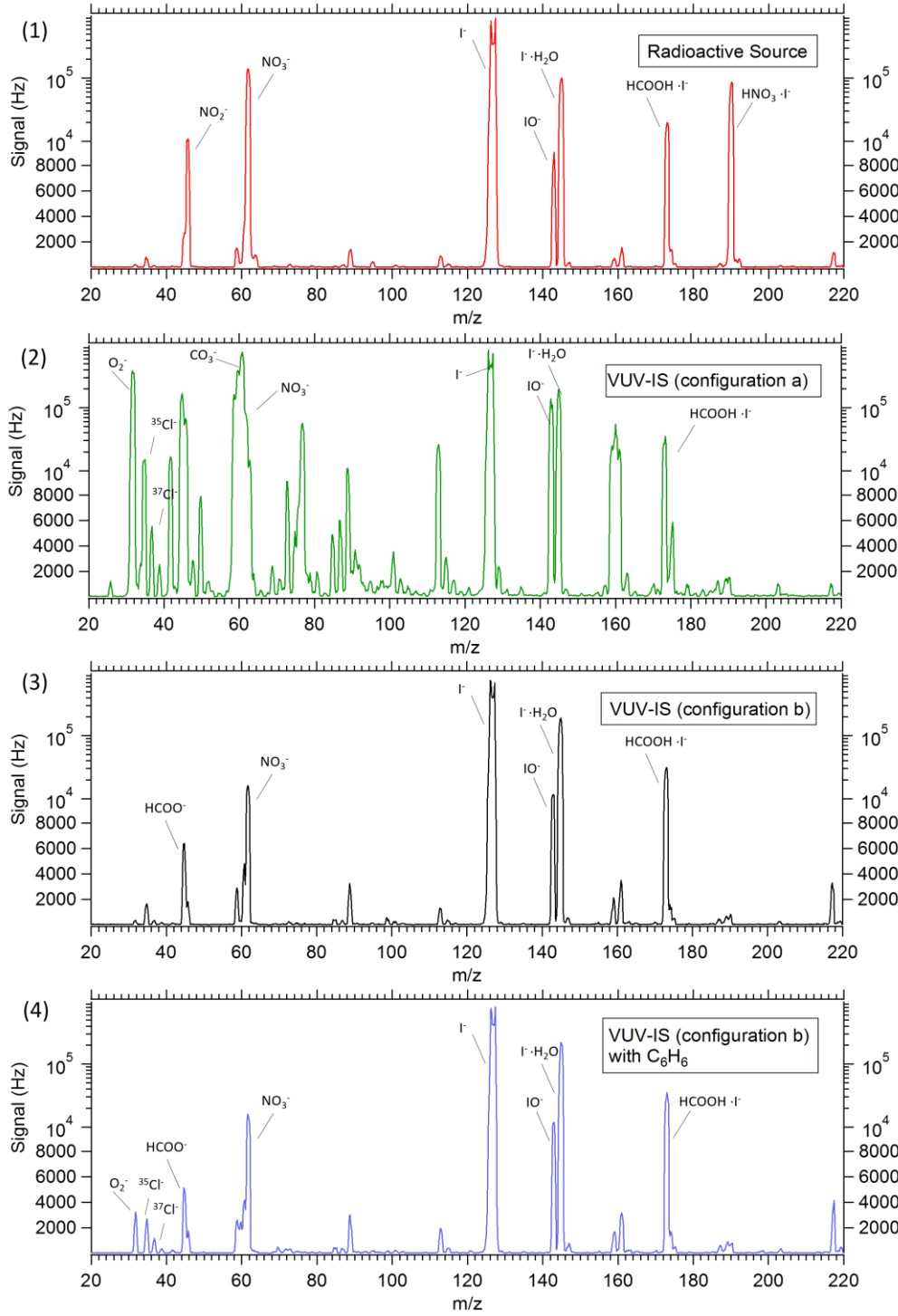

**Figure 5. Mass spectra of ambient air from a Q-CIMS with (1) a standard radioactive ion source, (2) VUV-IS in configuration (a), (3) VUV-IS in configuration (b), and (4) VUV-IS in configuration (b) with ~100 ppmv of $C_6H_6$ and ~10 ppmv of $CH_3I$. Note that the $I^-$ signal is saturated in all mass spectra.**

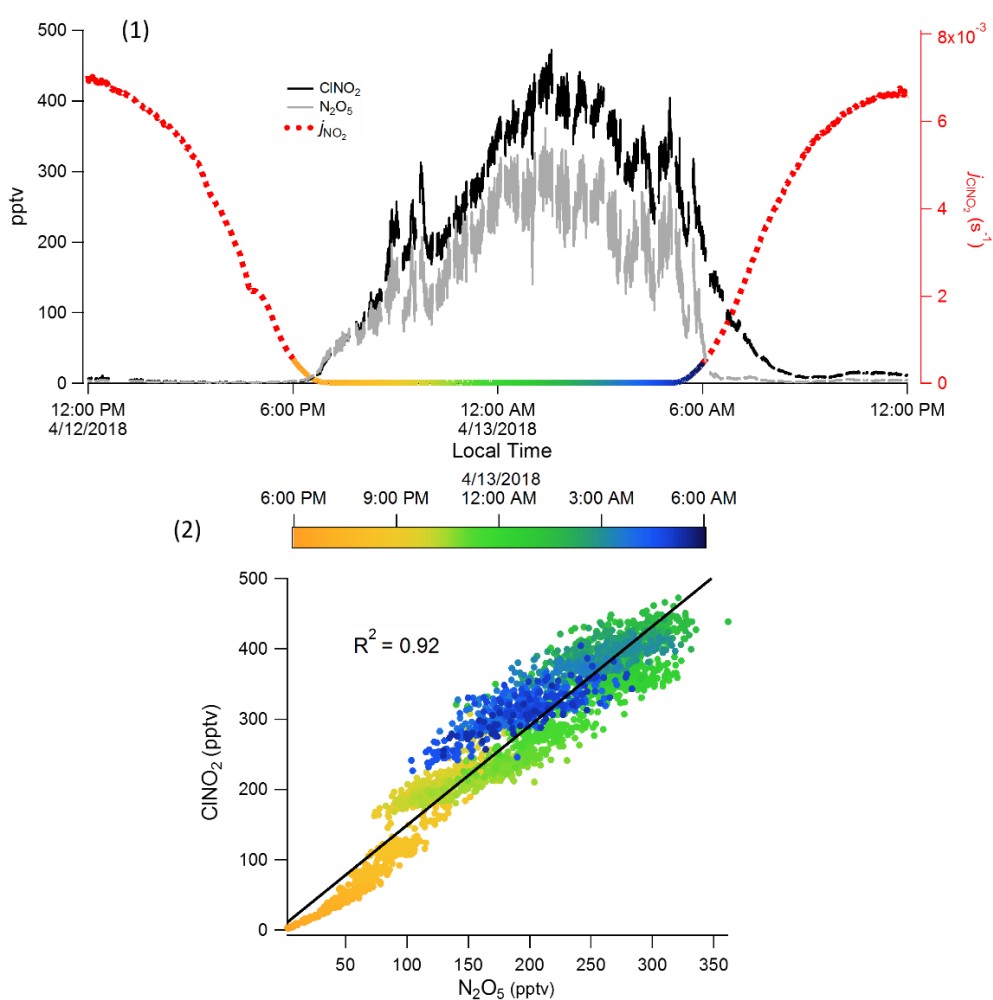

**Figure 6. Measurements of ClNO₂ and N₂O₅ using I⁻-CIMS with VUV-IS between 12 pm of April 12 and 12 pm of April 13, 2018 during OPECE campaign. (1) time series along with $j_{NO_2}$ (to delineate night and day) and (2) a correlation plot of ClNO₂ concentration versus N₂O₅ concentration during the night time (6 pm, April 12 to 6 am, April 13).**

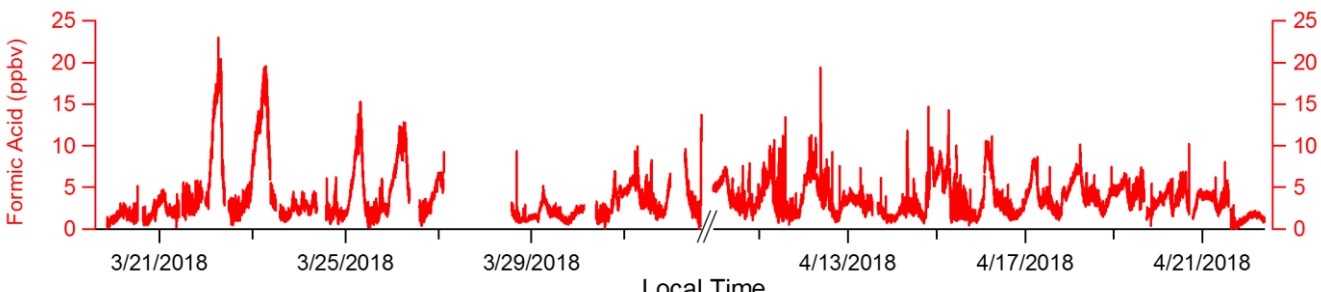

**Figure 7. Time series of ambient formic acid concentrations measured by I⁻-CIMS with VUV-IS from March 20 to April 22, 2018 during OPECE campaign.**

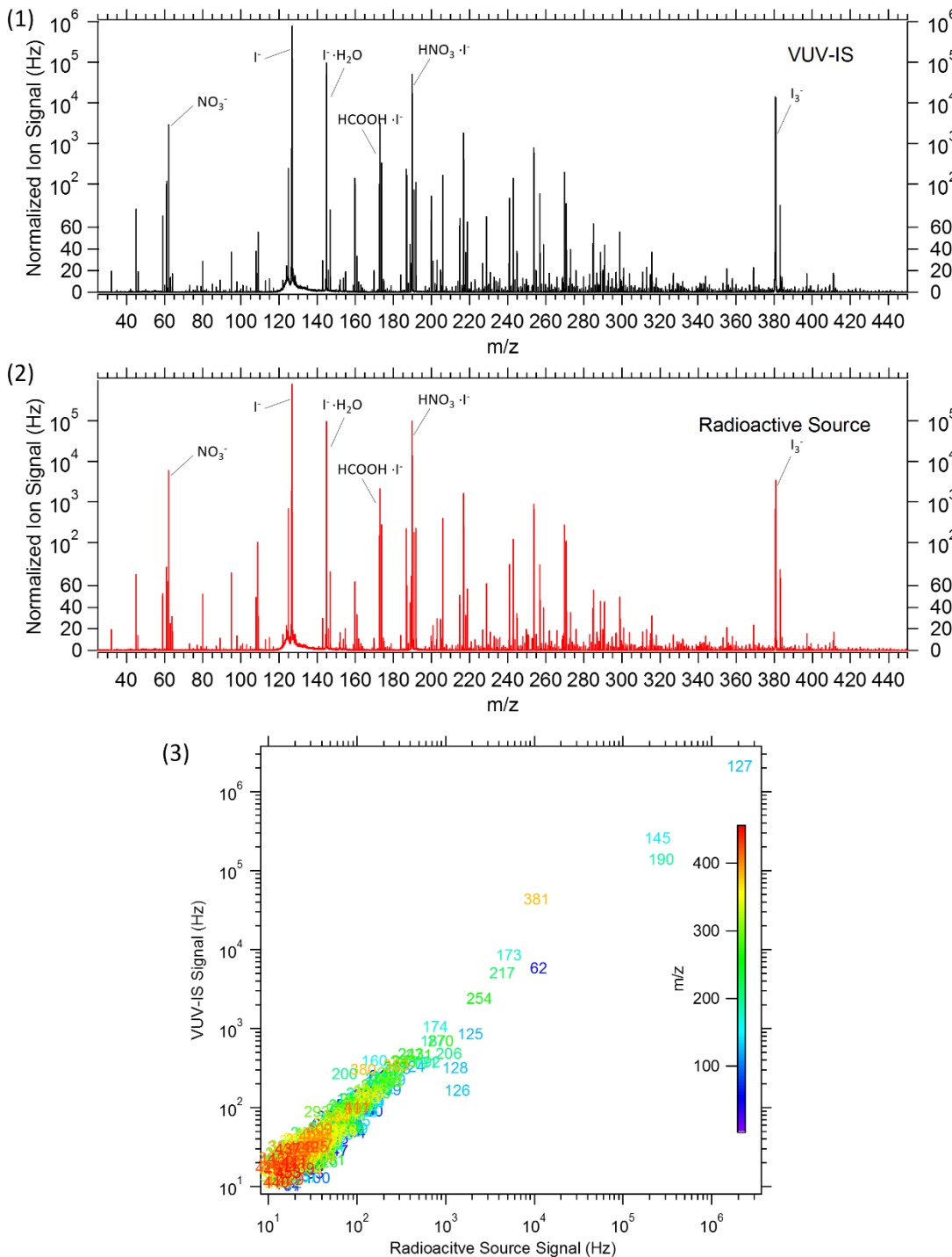

**Figure 8. Mass spectra of ambient air in Boulder, CO from a TOF-CIMS with (1) a VUV-IS (2) a radioactive source. The bottom panel (3) is a correlation plot of the individual mass signals with the VUV-IS versus those obtained with a radioactive source, with each ion labelled by its nominal mass.**

**Table 1: Experiment conditions, sensitivities and limits of detection (LODs) for VUV-IS configuration (b)**

| Exp. | | $CH_3I$ only | $CH_3I$ and $C_6H_6$ | $CH_3I$ only | $CH_3I$ and $C_6H_6$ |
|---|---|---|---|---|---|
| flow tube pressure (Torr) | | 20 | 20 | 40 | 40 |
| $CH_3I$ mixing ratio (ppmv) | | 8.8 – 86.5 | 8.8 | 1.8 – 19.0 | 1.8 |
| $C_6H_6$ mixing ratio (ppmv) | | 0 | 0 – 229.2 | 0 | 0 – 58.9 |
| formic acid | sensitivity (Hz pptv$^{-1}$) [a] | 79 - 147 | 79 - 158 | 77 - 128 | 77 - 157 |
| | 1 min LOD (pptv) [a, b] | 0.78 – 0.74 | 0.78 – 0.67 | 0.88 – 0.62 | 0.88 – 0.67 |
| $Cl_2$ | sensitivity (Hz pptv$^{-1}$) [a] | 83 – 161 | 83 - 157 | 79 - 149 | 79 - 166 |
| | 1 min LOD (pptv) [a, b] | 0.82 – 0.72 | 0.82 – 0.68 | 0.92 – 0.64 | 0.92 – 0.52 |
| $ClNO_2$ | sensitivity (Hz pptv$^{-1}$) [a] | 77 - 154 | 77 - 152 | 85 - 148 | 85 - 138 |
| | 1 min LOD (pptv) [a, b] | 0.45 – 0.24 | 0.45 – 0.31 | 0.43 – 0.17 | 0.43 – 0.19 |

[a] Sensitivities and detection limits are for $HCOOH \cdot I^-$ (m/z = 173 amu), $Cl_2 \cdot I^-$ (m/z = 197 amu) and $ClNO_2 \cdot I^-$ (m/z = 208 amu).

[b] Signal to noise ratio = 3:1

**Table 2: Experiment conditions, sensitivities and limits of detection (LODs) for VUV-IS configuration (a)**

| Exp. | | $CH_3I$ only | $CH_3I$ and $C_6H_6$ | $CH_3I$ only | $CH_3I$ and $C_6H_6$ |
|---|---|---|---|---|---|
| flow tube pressure (Torr) | | 20 | 20 | 40 | 40 |
| $CH_3I$ mixing ratio (ppmv) | | 9.6 - 83 | 9.6 | 1.6 – 19 | 1.6 |
| $C_6H_6$ mixing ratio (ppmv) | | 0 | 0 - 175 | 0 | 0 - 54 |
| formic acid | sensitivity (Hz pptv$^{-1}$) [a] | 228 - 761 | 225 - 761 | 171 - 591 | 171 - 703 |
| | 1 min LOD (pptv) [a, b] | 0.44 – 0.36 | 0.59 – 0.23 | 0.52 – 0.28 | 0.68 – 0.21 |
| $Cl_2$ | sensitivity (Hz pptv$^{-1}$) [a] | 212 – 754 | 210 - 740 | 173 - 605 | 173 - 694 |
| | 1 min LOD (pptv) [a, b] | 0.56 – 0.30 | 0.49 – 0.27 | 0.62 – 0.24 | 0.48 – 0.24 |
| $ClNO_2$ | sensitivity (Hz pptv$^{-1}$) [a] | 193 - 687 | 204 - 679 | 153 - 570 | 169 - 605 |
| | 1 min LOD (pptv) [a, b] | 0.22 – 0.12 | 0.20 – 0.10 | 0.07 – 0.05 | 0.12 – 0.07 |

[a] Sensitivities and detection limits are for $HCOOH \cdot I^-$ (m/z = 173 amu), $Cl_2 \cdot I^-$ (m/z = 197 amu) and $ClNO_2 \cdot I^-$ (m/z = 208 amu).

[b] Signal to noise ratio = 3:1