# Peer review of "A Vacuum Ultraviolet Ion Source (VUV-IS) for Iodide-Chemical Ionization Mass Spectrometry: A Substitute for Radioactive Ion Sources"

_Atmospheric Measurement Techniques, 2020_

## Referee Comment (RC1) · Joachim Franzke (Referee) · 4 Feb 2020

In their manuscript ac-2019-02044c "A Vacuum Ultraviolet Ion Source (VUV-IS) for Iodide-Chemical Ionization Mass Spectrometry: A Substitute for Radioactive Ion Sources" the authors describe how to apply a commercially available krypton lamp that emits two Kr lines in the VUV at 123.582 nm (10.030 eV) and 116.486 nm (10.641 eV). The VUV light photo-ionizes either methyl iodide or benzene to form cations and photoelectrons. The authors used two configurations to investigate the different sensitivities and to compare this ionisation with that of a radioactive source.

Here are some items which should be improved:

[Figure]

Line 17: the light, which is emitted are atomic lines as mentioned above and not wave-length bands as written by the authors. The right values for these lines are given above.

The description of figure 3 and figure 5 is sometimes a little confusing. This should be improved. Maybe the authors could indicate figure 3 a,b,c,d and 5 a,b,c,d instead of saying "upper left" and "lower left". At line 176 the authors write: " At 40 Torr, up to 58.9 ppmv $C_6H_6$ was added to 1.8 ppmv of $CH_3I$ to reach the maximum level of sensitivities (157, 166, and 138 Hz pptv-1 for formic acid, $Cl_2$ and $ClNO_2$, respectively) when using 19.0 ppmv of $CH_3I$." I suspect the 58.9 ppmv should be exchanged by 19 ppmv as shown in the lower left part of figure 5. Also in Table 1 this should be corrected: Instead of 0-58.9 it should be written 0-19, I guess. It would also be easier for the reader to follow, when in figure 3 upper left would be written in the graph: 9.6 ppmv $CH_3I$, in the lower left 1.6 ppmv $CH_3I$, in figure 5 upper left: 8.8 ppmv $CH_3I$ and lower left : 1.8 ppmv $CH_3I$.

In the figure caption the same values like in the text should be used: 110 ppmv and 8.8 ppmv. It would also be good for the reader to indicate some masses of the spectra like it is explained in the text.

The authors used two configurations to show that configuration a generates many additional ions compared to a radioactive source. The explanation is that these ions might be created by the VUV radiation. Do the authors see all these additional ions also when they do not use any methyl iodine or benzene?

When the above mentioned items are corrected or improved the manuscript should be published.

---

## Referee Comment (RC2) · Anonymous Referee #2 · 13 Feb 2020

Ji et al present experimental data pertaining to the characterisation and deployment of a VUV ion-source using a commercially available photo-ionisation lamp that provides a viable alternative to use of polonium in an I-CIMS, at least for some trace-gases.

The VUV source appears to provide sensitivities for detection of HC(O)OH, ClNO2 and Cl2 that are comparable to those achieved with Po210, but without the logistic disadvantages related to deployment of radioactive sources in field measurements. A further advantage is that, unlike Po210, the VUV source does not generate HNO3.

The manuscript is short and to the point, which is fine; but its organisation could be substantially improved and there are many minor errors/omissions that need attention.

A general point. I- CIMS was first deployed for measurement of PAN and this method is still widely used. Why was PAN not tested in the lab ? It appears from fig. 4 that the signal at 59 amu is much larger when using the VUV (config a) compared to Po210. Is this due to a better sensitivity, a worse background or changing ambient conditions. If the latter is true, it is not clear what this figure hopes to convey. As mentioned below, a spectra obtained in zero-air should be presented in order to assess the potential of the VUV-source properly.

L23 How does deploying the instrument in the field for a month demonstrate reliability? Section 3.4 and Fig 7 do not really help convince (beyond simply stating that it was operated continuously for a month) that the VUV source can be operated under field conditions. The data (HC(O)OH measurements) are not discussed in terms of their quality or compared to other measurements. Did the primary ion-current (lamp output) change over one month of operation? What does the manufacturer state concerning the operational lifetime (in hours) of the PKS 106?

L26 The spectrum in ambient air was cleaner. This could mean that the VUV source does not detect everything the Po210 source does. Alternatively, it could imply that the primary ion spectrum is less complex. Please clarify. Again, it would be nice to see a comparison of the primary ion spectra in zero air, at least in the supplement.

L34 ...used to detect many atmospheric trace gases.... But then only BrO and PAN are named. Why not simply list what classes of trace gases have been measured (organic, halogen, nitrates etc etc)?

L43 "little" = "few" ?

L60 ...which can be attached by CH3I... = which can attach to CH3I....?

L61 "Benzene has a larger cross-section (than CH3I). Please quote the cross-sections (and wavelengths) for benzene and CH3I here as well as the ionisation potentials (or refer to a section where they are listed/tabulated)

L78/80 The acronyms Q-CIMS and I- -CIMS appear to be used randomly. In Fig. 1 the term I–Q-CIMS is also used. Please unify throughout the manuscript.

L103 Some details pertaining to the detection of Cl2 (a reference to the instrument, total uncertainty, cross-section used etc) would be appropriate here.

L110 Was the DC power supply a commercial one (also from Heraeus ?)

L134 The terms IMR and flow-tube (eg. In Fig.1) are used interchangeably. Please stick to one. As it is not strictly a "flow-tube" Ion Molecule Reactor might be the better choice.

L144 What is special about the photoelectrons generated on the metal surfaces. Explain why they are a problem ? (Presumably because they can attach to O2 in this part of the instrument). On the other hand, could metal surfaces in N2 / CH3I gas provide a useful source of photo-electrons upstream of the IMR ?

Section 3, Results. The first paragraph in the results section summarises all of the results and introduces all of the Figures from the laboratory tests. As the same conclusions are made in the individual sections that follow (but with the underlying data to support them) this paragraph is redundant. I suggest it can be removed and integrated with the conclusions.

L158 Are the mixing ratios of CH3I those in the ion-source-flow or in the IMR (after dilution) ? Please make this clear throughout the manuscript.

L199 I'm surprised that the authors did not add SF6 to test the potential of extension to the use of the VUV-IS. This would have been a very simple experiment. I would encourage the authors to do this "10 minute experiment" and add the qualitative observation (formation of SF6- ?) to this manuscript. Are other lamps available (different noble gases and emission energies) to extend the use of photo-ionisation sources to generate other primary ions for CIMS. Or, put another why, why choose a Krypton line ?

L200 "electron attaching compounds" = trace gases with large electron affinities ?

Section 3.4 I'm not sure what this section seeks to achieve. The information that the lamp can be used for 18 months without degradation of the signal is important but was not gleaned from a 1 month campaign and is lost in this section. I suggest adding this information to the methods section, perhaps with manufactures data about the lifetime of the lamp. Also "no obvious degradation of signal" could be presented in a more quantitative manner. What "signal" is referred to here ? What information (Fig. 7) do the J-NO2 values convey? Seeing that there is no discussion of the data, th J-values simply delineate night and day. But what does that tell us ? Why were HC(O)OH mixing ratios chosen to illustrate that the ion-source worked ? Was nothing else measured ? Why do the formic acid mixing ratios maximise at nighttime (expected ?).

I recommend removing this section completely unless significantly more use is made of the field data (a comparison with another instrument would e.g. have been useful).

Section 4. The "discussion" is weak. I suggest taking this text and combining it with the (weak) "conclusions" text to generate a section "discussion of results and conclusions" or something like that.

L255 Extend the conclusions by taking the first paragraph of text from the results section.

Fig 4 (1) and (3) It would be good to add the ions to the figures. The HNO3I- ion at 1190 amu is the main difference between the Po210 and VUV ion sources and this would be nicely highlighted if the ion-peaks were labelled.

In this context, what is the ion at $\sim$ 144 amu (next to I-H2O)? and at $\sim$ 170 amu ?

SI: The cross-section of CH3I is x 10-17 (not e17). Give a citation for the cross-section.
* * *

---

## Author Comment (AC2) · 22 Apr 2020

Referee comment: "A general point. I- CIMS was first deployed for measurement of PAN and this method is still widely used. Why was PAN not tested in the lab? It appears from fig. 4 that the signal at 59 amu is much larger when using the VUV (config a) compared to Po210. Is this due to a better sensitivity, a worse background or changing ambient conditions? If the latter is true, it is not clear what this figure hopes to convey. As mentioned below, a spectra obtained in zero-air should be presented in order to assess the potential of the VUV-source properly."

Author response: We chose to focus on an unheated I–CIMS for this work as this is

probably the most common CIMS method used in the community. The PAN detection by TD-CIMS is more specialized, involves a heated inlet, and is not as widely used. The mass spectrum with the VUV source in configuration (a) does have an obvious background at 59 amu as well as other masses. We showed this mass spectrum to demonstrate the issue of stray photoelectrons which would impact many measurements. We also showed simple solutions to minimize this issue. We have added a mass spectrum to the SI (Figure S3) to show that PAN can be detected with TD-CIMS using a VUV-IS. We have also a short section in the results to report the quick tests requested by the reviewer.

Referee comment: "L23 How does deploying the instrument in the field for a month demonstrate reliability? Section 3.4 and Fig 7 do not really help convince (beyond simply stating that it was operated continuously for a month) that the VUV source can be operated under field conditions. The data (HC(O)OH measurements) are not discussed in terms of their quality or compared to other measurements. Did the primary ion-current (lamp output) change over one month of operation? What does the manufacturer state concerning the operational lifetime (in hours) of the PKS 106?"

Author response: Running the CIMS with a lamp for a month in a remote location does demonstrate reliability but we agree it doesn't demonstrate the quality of the measurement. Our primary goal for this CIMS during the OPECE campaign was to measure halogen containing compounds. However, we found the presence of halogens to be intermittent so we chose to show the formic acid data. Formic acid is routinely measured by I- CIMS and is ubiquitous in the atmosphere as both an emission and a secondary chemical product. So the formic data offers a way to show the CIMS with a lamp could be operated continuously for an extended period. This was one of our major concerns going into the field campaign as we were deploying in an unheated shelter in a remote location in a bird sanctuary in the Yellow River Delta with intermittent power issues. For this reason, we have provided more details of the physical setup and describe the temperature variance experienced by the CIMS.

In order to address the data quality question, we have added a figure with representative observations of ClNO2 and N2O5 for this site when halogens were observed (Figure 6). This figure is consistent with expected behavior of ClNO2 and N2O5 as they accumulate during night time and decay after sunrise. This indicates the performance of the CIMS is reasonable.

The lifetime of this lamp is claimed to be 4000 hours (∼5.5 months) by the manufacturer, and we did not observe an obvious degradation of CIMS sensitivity. The CIMS sensitivity to formic acid was measured to be 185.2 ± 48.3 Hz pptv-1 during the first 24 hours, and 180.5 ± 24.3 Hz pptv-1 during the last 24 hours of the mission. So we did not observe any drop in sensitivity that could be attributed to a decrease in light intensity from the lamp. We have reported the starting and ending sensitivity in the results section.

Referee comment: "L26 The spectrum in ambient air was cleaner. This could mean that the VUV source does not detect everything the Po210 source does. Alternatively, it could imply that the primary ion spectrum is less complex. Please clarify. Again, it would be nice to see a comparison of the primary ion spectra in zero air, at least in the supplement."

Author response: Both the VUV and polonium sources produce similar amounts of I- as well as other ions. So it is hard to understand why the VUV source does not detect everything in the same manner as the polonium source. They both utilize reactions of I-. We can certainly obtain a mass spectra in zero air, although we note that most complications with CIMS come from the presence of ozone and water, but we have trouble accessing our laboratory at this time due to COVID.

Referee comment: "L34 . . . used to detect many atmospheric trace gases. . . .. But then only BrO and PAN are named. Why not simply list what classes of trace gases have been measured (organic, halogen, nitrates etc)?"

Author response: We have modified the texts as suggested.

Referee comment: "L43 "little" = "few"?"

Author response: We have modified the texts as suggested.

Referee comment: "L60 . . .which can be attached by CH3I. . . = which can attach to CH3I. . ..?"

Author response: We have modified the texts as suggested.

Referee comment: "L78/80 Benzene has a larger cross-section (than CH3I). Please quote the cross-sections (and wavelengths) for benzene and CH3I here as well as the ionisation potentials (or refer to a section where they are listed/tabulated)"

Author response: This values was quoted in line 65.

Referee comment: "L103 Some details pertaining to the detection of Cl2 (a reference to the instrument, total uncertainty, cross-section used etc) would be appropriate here."

Author response: We have provided more detailed information and reference in the text.

Referee comment: "L110 Was the DC power supply a commercial one (also from Heraeus?)"

Author response: We have added the model and manufacturer.

Referee comment: "L134 The terms IMR and flow-tube (eg. In Fig.1) are used interchangeably. Please stick to one. As it is not strictly a "flow-tube" Ion Molecule Reactor might be the better choice."

Author response: The terms of IMR is more commonly used when talking about Aerodyne TOF-CIMS, and "flow tube" are more commonly used when talking about quadrupole CIMS, so we prefer to stick to the convention to avoid confusion to readers.

Referee comment: "L144 What is special about the photoelectrons generated on the metal surfaces. Explain why they are a problem? (Presumably because they can attach

to O2 in this part of the instrument). On the other hand, could metal surfaces in N2 / CH3I gas provide a useful source of photo-electrons upstream of the IMR?"

Author response: The photoelectrons generated in the flow tube are exposed to O2 which electron attaches to make O2- and leads to a series of interfering masses. Our preliminary tests shows that using metal surfaces can produce only about 1% I- of that achieved with using a gas phase absorber (CH3I or C6H6).

Referee comment: "Section 3, Results. The first paragraph in the results section summarises all of the results and introduces all of the Figures from the laboratory tests. As the same conclusions are made in the individual sections that follow (but with the underlying data to support them) this paragraph is redundant. I suggest it can be removed and integrated with the conclusions."

Author response:

Good comment – We have moved this to the end of the paper and used as the first paragraph of a summary and conclusion section.

Referee comment: "L158 Are the mixing ratios of CH3I those in the ion-source-flow or in the IMR (after dilution)? Please make this clear throughout the manuscript."

Author response: All the mixing ratios of CH3I and benzene in the manuscripts are those in the ion source flow. We mentioned this in line 151 for TOF-CIMS and line 159, 201 for Q-CIMS. We have also clarified it in line 144.

Referee comment: "L199 I'm surprised that the authors did not add SF6 to test the potential of extension to the use of the VUV-IS. This would have been a very simple experiment. I would encourage the authors to do this "10 minute experiment" and add the qualitative observation (formation of SF6- ?) to this manuscript."

Author response: We are working on using SF6- with the lamp. This is really not a "10 minute experiment", because SF6-chemistry is more complicated than I- chemistry. E.g. its reaction with water vapor and O3 are troublesome to both its selectivity and

stability, so we have to minimize the reaction time and number density in the flow tube. So we don't have extensive or conclusive results at this time. We have added a figure (S4) that shows preliminary measurements of formic and acetic acid using an SF6- during test flights.

Referee comment: "Are other lamps available (different noble gases and emission energies) to extend the use of photo-ionisation sources to generate other primary ions for CIMS. Or, put another why, why choose a Krypton line?"

Author response: Photoionization detector lamps (PID) are available with a variety of gas fills including argon, krypton and xenon gas. However xenon's photon energy (9.6 eV) is lower than krypton (10 and 10.6 eV). The argon lamp has a higher photo energy (11.8 eV) but it also lithium fluoride (LiF) as a window material which is hygroscopic. The krypton lamp uses a magnesium Fluoride (MgF2) window which is less hygroscopic.

Referee comment: "L200 "electron attaching compounds" = trace gases with large electron affinities?"

Author response: Electron attaching compound are molecules like O2, SF6, CH3I that attach electrons to make species such as O2-, SF6-, I-. However, SF6 and O2 don't have particularly large electron affinities and the electron affinity of CH3I is a more difficult concept as it dissociative attaches to form I-, so we prefer electron attaching compounds,

Referee comment: "Section 3.4 I'm not sure what this section seeks to achieve. The information that the lamp can be used for 18 months without degradation of the signal is important but was not gleaned from a 1 month campaign and is lost in this section. I suggest adding this information to the methods section, perhaps with manufactures data about the lifetime of the lamp. Also "no obvious degradation of signal" could be presented in a more quantitative manner. What "signal" is referred to here? What information (Fig. 7) do the J-NO2 values convey? Seeing that there is no discussion of

the data, th J-values simply delineate night and day. But what does that tell us? Why were HC(O)OH mixing ratios chosen to illustrate that the ion-source worked ? Was nothing else measured? Why do the formic acid mixing ratios maximise at nighttime (expected?). I recommend removing this section completely unless significantly more use is made of the field data (a comparison with another instrument would e.g. have been useful)."

Author response: We have been using the same VUV-IS since the OPECE field mission (Spring 2018) till 2020 for both lab studies and field measurements, and we have not find significant degradation of CIMS sensitivity. As mentioned before, the krypton lamp's lifetime is expected to ~5.5 month of continuous use

Yes the JNO2 is to simply delineate night and day. As mentioned above, Figure 7 is to show that the VUV-IS was operated continuously for a month. We are choosing formic acid because it is ubiquitous and easy to measure and calibrate. We are not claiming anything about the quality of the formic acid measurements. As discussed before, we also presented our measurements on ClNO2 and N2O5 as a support of our data quality (Figure 6).

Referee comment: "Section 4. The "discussion" is weak. I suggest taking this text and combining it with the (weak) "conclusions" text to generate a section "discussion of results and conclusions" or something like that."

Author response: We have modified the text by moving the intro paragraph to an end summary and conclusion sections. We also feel that we have made a strong conclusion – The VUV lamp can eliminate the use of radioactivity for CIMS.

Referee comment: "L255 Extend the conclusions by taking the first paragraph of text from the results section."

Author response: We have moved the summary paragraph to the conclusion and make it a "summary and conclusion section".

Referee comment: "Fig 4 (1) and (3). It would be good to add the ions to the figures. The HNO3I- ion at 1190 amu is the main difference between the Po210 and VUV ion sources and this would be nicely highlighted if the ion-peaks were labelled. In this context, what is the ion at âĹij 144 amu (next to I-H2O)? and at âĹij 170 amu?"

Author response: We have added labels to the larger signal ions in the spectra. Please refer to Figure 5 and 8 above. There is no peak at 144 but there is a peak at 143 amu that corresponds to IO-. The peak at $\sim$ 170 corresponds to HCOOH·I- at m/z 173.

Referee comment: "SI: The cross-section of CH3I is x 10-17 (not e17). Give a citation for the cross-section."

Author response: We have modified the texts as suggested.
* * *

---

## Author Response (AR1)

We greatly value the careful reading and the detailed comments provided by the referees. The responses to the comments of the referees in our direct reply (shown below) and within the revised manuscript (see marked copy) are provided below. The pages and lines indicated below correspond to those in the marked copy.

**Response to RC1 (Referees' comments are italicized)**

**1. Referee comment:** *"... the authors describe how to apply a commercially available krypton lamp that emits two Kr lines in the VUV at 123.582 nm (10.030 eV) and 116.486 nm (10.641 eV)." "Line 17: the light, which is emitted are atomic lines as mentioned above and not wavelength bands as written by the authors. The right values for these lines are given above."*

**Author response:**
We have modified the texts as suggested. However, we note that the emission lines from krypton lamps exhibit significant pressure broadening.

**Page 1 line 16: "The VUV-IS utilizes a compact krypton lamp that emits light at two wavelengths corresponding to energies of ~10.030 and 10.641 eV."**

**2. Referee comment:** *"The description of figure 3 and figure 5 is sometimes a little confusing. This should be improved. Maybe the authors could indicate figure 3 a,b,c,d and 5 a,b,c,d instead of saying "upper left" and "lower left". "*

**Author response:**
We modified Figure 3 and 5 (Figure 4 and 3 in revised manuscript, respectively) "upper left" etc to i, ii, iii, iv, to avoid confusions with VUV-IS configuration (a) and (b).

**3. Referee comment:** *"At line 176 the authors write: "At 40 Torr, up to 58.9 ppmv C6H6 was added to 1.8 ppmv of CH3I to reach the maximum level of sensitivities (157, 166, and 138 Hz pptv-1 for formic acid, Cl2 and ClNO2, respectively) when using 19.0 ppmv of CH3I." I suspect the 58.9 ppmv should be exchanged by 19 ppmv as shown in the lower left part of figure 5. Also in Table 1 this should be corrected: Instead of 0-58.9 it should be written 0-19, I guess. It would also be easier for the reader to follow, when in figure 3 upper left would be written in the graph: 9.6 ppmv CH3I, in the lower left 1.6 ppmv CH3I, in figure 5 upper left: 8.8 ppmv CH3I and lower left : 1.8 ppmv CH3I. In the figure caption the same values like in the text should be used: 110 ppmv and 8.8ppmv."*

**Author response:**
The text in line 176 (line 187 in revised manuscript) and Table 1 were both correct (58.9 ppmv of benzene was used here). The lower left panel (iv) of Figure 5 (Figure 3 in revised manuscript) was modified to the correct plot. The amount of $CH_3I$ used was provided on the figures and captions as suggested.

[Figure]

**Figure 3. Q-CIMS using VUV-IS configuration (b): (i) sensitivity as a function of CH₃I at 20 Torr. (ii) sensitivity as a function of C₆H₆ at 20 Torr with 8.8 ppmv of CH₃I. (iii) sensitivity as a function of CH₃I at 40 Torr. (iv) sensitivity as a function of C₆H₆ at 40 Torr with 1.8 ppmv of CH₃I.**

[Figure]

**Figure 4. Q-CIMS using VUV-IS configuration (a): (i) sensitivity as a function of CH₃I at 20 Torr. (ii) sensitivity as a function of C₆H₆ at 20 Torr with 9.6 ppmv of CH₃I. (iii) sensitivity as a function of CH₃I at 40 Torr. (iv) sensitivity as a function of C₆H₆ at 40 Torr with 1.6 ppmv of CH₃I.**

**4. Referee comment:** *"It would also be good for the reader to indicate some masses of the spectra like it is explained in the text."*

**Author response:**
We have modified Figure 5 and 8 as suggested.

[Figure]

**Figure 5. Mass spectra of ambient air from a Q-CIMS with (1) a standard radioactive ion source, (2) VUV-IS in configuration (a), (3) VUV-IS in configuration (b), and (4) VUV-IS in configuration (b) with ~100 ppmv of C₆H₆ and ~10 ppmv of CH₃I. Note that the I⁻ signal is saturated in all mass spectra.**

[Figure]

**Figure 8.** Mass spectra of ambient air in Boulder, CO from a TOF-CIMS with (1) a VUV-IS (2) a radioactive source. The bottom panel (3) is a correlation plot of the individual mass signals with the VUV-IS versus those obtained with a radioactive source, with each ion labeled by its nominal mass.

**5. Referee comment:** *"The authors used two configurations to show that configuration a generates many additional ions compared to a radioactive source. The explanation is that these*

*ions might be created by the VUV radiation. Do the authors see all these additional ions also when they do not use any methyl iodine or benzene?"*

**Author response:**
Yes we did tests without $CH_3I$ or benzene, and additional peaks (e.g. $Cl^-$, $O_2^-$ and masses associated with $O_2^-$ chemistry) similar to those in Figure 5(2) where observed when using VUV-IS configuration (a). These additional peaks didn't show up when using VUV-IS configuration (b). These tests also indicate the source of the interference peaks are photoelectrons generated on the illuminated surface of the flow tube.

**Response to RC2 (Referees' comments are italicized)**

**6. Referee comment:** *"A general point. I- CIMS was first deployed for measurement of PAN and this method is still widely used. Why was PAN not tested in the lab? It appears from fig. 4 that the signal at 59 amu is much larger when using the VUV (config a) compared to Po210. Is this due to a better sensitivity, a worse background or changing ambient conditions? If the latter is true, it is not clear what this figure hopes to convey. As mentioned below, a spectra obtained in zero-air should be presented in order to assess the potential of the VUV-source properly."*

**Author response:**
We chose to focus on an unheated $I^-$-CIMS for this work as this is probably the most common CIMS method used in the community. The PAN detection by TD-CIMS is more specialized, involves a heated inlet, and is not as widely used. The mass spectrum with the VUV source in configuration (a) does have an obvious background at 59 amu as well as other masses. We showed this mass spectrum to demonstrate the issue of stray photoelectrons which would impact many measurements. We also showed simple solutions to minimize this issue. We have added a mass spectrum to the SI (Figure S3) to show that PAN can be detected with TD-CIMS using a VUV-IS. We have also a short section in the results to report the quick tests requested by the reviewer.

**Page 5 line 150: "**
**2.5 TD-CIMS**
**The sensitivity of TD-CIMS with a VUV-IS (Slusher et al., 2004) was also tested for PAN. The configuration of the TD-CIMS system used in this work is almost identical to that described in Lee et al. (2019) with the radioactive source replaced with the VUV-IS in configuration (b). A known amount of PAN was generated using a photolytic source similar to that described by Warneck and Zerbach (1992). A calibration standard of 1 ppbv of PAN was produced by adding the output of the photolytic source to PAN free ambient air. PAN free air was generated by passing ambient air through a QF 40 nipple filled with stainless steel wool heated to 150 °C (Flocke et al., 2005)."**

**Page 9 line 253: "**
**3.6 PAN Measurement Tests**
**Preliminary tests using the Q-CIMS as a TD-CIMS demonstrated the potential of the VUV-IS for use in the measurement of PAN. Mass spectra of ambient air with and without**

**PAN are shown in Figure S3. The sensitivity towards PAN was observed to be 49.4 Hz/pptv with an LOD (Signal to noise ratio = 3:1) of 0.64 pptv for a 1 minute integration. No significant interferences were observed during the ambient air tests."**

**References:**

**Warneck, P. and Zerbach, T.: Synthesis of peroxyacetyl nitrate in air by acetone photolysis, Environ. Sci. Technol., 26, 74– 79, 1992.**

**Flocke, F. M., Weinheimer, A. J., Swanson, A. L., Roberts, J. M., Schmitt, R., and Shertz, S.: On the measurement of PANs by gas chromatography and electron capture detection, J. Atmos. Chem., 52, 19–43, doi:10.1007/s10874-005-6772-0, 2005.**

[Figure]

**Figure S3. Mass spectra of zeroed ambient air with and without PAN calibration standard.**

**7. Referee comment:** *"L23 How does deploying the instrument in the field for a month demonstrate reliability? Section 3.4 and Fig 7 do not really help convince (beyond simply stating that it was operated continuously for a month) that the VUV source can be operated under field conditions. The data (HC(O)OH measurements) are not discussed in terms of their quality or compared to other measurements. Did the primary ion-current (lamp output) change over one month of operation? What does the manufacturer state concerning the operational lifetime (in hours) of the PKS 106?"*

**Author response:**
Running the CIMS with a lamp for a month in a remote location does demonstrate reliability but we agree it doesn't demonstrate the quality of the measurement. Our primary goal for this CIMS during the OPECE campaign was to measure halogen containing compounds. However, we found the presence of halogens to be intermittent so we chose to show the formic acid data. Formic acid is routinely measured by I- CIMS and is ubiquitous in the atmosphere as both an emission and a secondary chemical product. So the formic data offers a way to show the CIMS with a lamp could be operated continuously for an extended period.

**Page 8 line 230: "Formic acid which is routinely measured by I⁻-CIMS and ubiquitous in the atmosphere as both an emission and a secondary chemical product was also monitored**

**during the campaign. These observations demonstrate that I⁻-CIMS with a VUV-IS could be operated continuously for an extended period."**

This was one of our major concerns going into the field campaign as we were deploying in an unheated shelter in a remote location in a bird sanctuary in the Yellow River Delta with intermittent power issues. For this reason, we have provided more details of the physical setup and describe the temperature variance experienced by the CIMS.

**Page 7 line 216: "The I⁻-CIMS was deployed in a shelter, with neither heating nor air conditioning, in a remote location in a bird sanctuary in the Yellow River Delta. The site experienced intermittent power interruptions and large ambient temperature variations, from -2.5 to 29.1 °C, with the temperature inside the shelter ranging from ~10 to 40 °C."**

In order to address the data quality question, we have added a figure with representative observations of $ClNO_2$ and $N_2O_5$ for this site when halogens were observed (Figure 6). This figure is consistent with expected behavior of $ClNO_2$ and $N_2O_5$ as they accumulate during night time and decay after sunrise. This indicates the performance of the CIMS is reasonable.

**Page 8 line 220: "The primary goal for the I⁻-CIMS during the OPECE campaign was to measure halogen containing compounds. However, we found the presence of halogens to be intermittent.  Figure 6 shows representative observations of $ClNO_2$ and $N_2O_5$ for this site when halogens were observed. This figure is consistent with the expected behavior of $ClNO_2$ and $N_2O_5$, both accumulate during night time, and both decrease after sunrise due to photolysis of $ClNO_2$ and thermal decomposition of $N_2O_5$ followed by photolysis of $NO_3$. $ClNO_2$ is a product of reaction between $N_2O_5$ and chloride containing aerosol (R1, Finlayson-Pitts et al., 1989), and $ClNO_2$ and $N_2O_5$ are well correlated ($R^2 = 0.94$, Figure 6(2)) during the night (6 pm, April 12 to 6 am, April 13). These measurements of $ClNO_2$ and $N_2O_5$ indicate the performance of the I⁻-CIMS with the VUV-IS is sufficient to capture atmospheric levels and variability.**

$$N_2O_5 + Cl^-_{(aq)} \rightarrow ClNO_2 + NO_3^-_{(aq)} \qquad \qquad \textbf{(R1)}$$ **"**

[Figure]

**Figure 6. Measurements of ClNO₂ and N₂O₅ using I⁻-CIMS with VUV-IS between 12 pm of April 12 and 12 pm of April 13, 2018 during OPECE campaign. (1) time series along with $j_{NO_2}$ (to delineate night and day) and (2) a correlation plot of ClNO₂ concentration versus N₂O₅ concentration during the night time (6 pm, April 12 to 6 am, April 13).**

The lifetime of this lamp is claimed to be 4000 hours (~5.5 months) by the manufacturer, and we did not observe an obvious degradation of CIMS sensitivity. The CIMS sensitivity to formic acid was measured to be 185.2 ± 48.3 Hz pptv⁻¹ during the first 24 hours, and 180.5 ± 24.3 Hz pptv⁻¹ during the last 24 hours of the mission. So we did not notice any drop in sensitivity that could be attributed to a decrease in light intensity from the lamp.

**Page 4 line 118: "The lifetime of this lamp is estimated to be 4000 hours (~5.5 months of continuous use) by the manufacturer."**

**Page 8 line 236: "Online calibration of formic acid was performed every 30 minutes during the mission. The CIMS sensitivity to formic acid was measured to be 185.2 ± 48.3 Hz pptv⁻¹ during the first day, and 180.5 ± 24.3 Hz pptv⁻¹ a month later, so we did not notice any drop in sensitivity that could be attributed to a decrease in light intensity from the lamp. In addition, we have used the same VUV-IS since the OPECE field mission (Spring 2018) through early 2020 for both lab studies and field measurements, and have not found obvious sensitivity degradations."**

[Figure]

**Figure S4. (1) Time series of formic acid signal (HCOO⁻ ·HF, m/z 65, red line), acetic acid signal (CH₃COO⁻ ·HF, m/z 79, black line), and ambient pressure (blue line). (2) A correlation plot of the CH₃COO⁻ ·HF signal (m/z 79) versus HCOO⁻ ·HF signal (m/z 65). Data was taken from the NCAR GV during a test flight based out of Broomfield, CO using a VUV-IS.**

**20. Referee comment:** *"Are other lamps available (different noble gases and emission energies) to extend the use of photo-ionisation sources to generate other primary ions for CIMS. Or, put another why, why choose a Krypton line?"*

**Author response:**
Photoionization detector lamps (PID) are available with a variety of gas fills including argon, krypton and xenon gas. However xenon's photon energy (9.6 eV) is lower than krypton (10 and 10.6 eV). The argon lamp has a higher photo energy (11.8 eV) but it also lithium fluoride (LiF) as a window material which is hygroscopic. The krypton lamp uses a magnesium Fluoride (MgF2) window which is less hygroscopic.

**21. Referee comment:** *"L200 "electron attaching compounds" = trace gases with large electron affinities?"*

**Author response:**
Electron attaching compound are molecules like $O_2$, $SF_6$, $CH_3I$ that attach electrons to make species such as $O_2^-$, $SF_6^-$, $I^-$. However, $SF_6$ and $O_2$ don't have particularly large electron affinities and the electron affinity of $CH_3I$ is a more difficult concept as it dissociative attaches to form $I^-$, so we prefer electron attaching compounds,

**22. Referee comment:** *"Section 3.4 I'm not sure what this section seeks to achieve. The information that the lamp can be used for 18 months without degradation of the signal is important but was not gleaned from a 1 month campaign and is lost in this section. I suggest adding this information to the methods section, perhaps with manufactures data about the lifetime of the lamp. Also "no obvious degradation of signal" could be presented in a more quantitative manner. What "signal" is referred to here? What information (Fig. 7) do the J-NO2 values convey? Seeing that there is no discussion of the data, th J-values simply delineate night and day. But what does that tell us? Why were HC(O)OH mixing ratios chosen to illustrate that the ion-source worked ? Was nothing else measured? Why do the formic acid mixing ratios maximise at nighttime (expected?). I recommend removing this section completely unless significantly more use is made of the field data (a comparison with another instrument would e.g. have been useful)."*

**Author response:**
We have been using the same VUV-IS since the OPECE field mission (Spring 2018) till 2020 for both lab studies and field measurements, and we have not find significant degradation of CIMS sensitivity. As mentioned in our response to comment 7, the krypton lamp's lifetime is expected to ~5.5 month of continuous use.

Yes the $JNO_2$ is to simply delineate night and day. As mentioned above, Figure 7 is to show that the VUV-IS was operated continuously for a month. We are choosing formic acid because it is ubiquitous and easy to measure and calibrate. We are not claiming anything about the quality of the formic acid measurements. As discussed in our response to comment 7, we also presented our measurements on $ClNO_2$ and $N_2O_5$ as a support of our data quality (Figure 6).

**23. Referee comment:** *"Section 4. The "discussion" is weak. I suggest taking this text and combining it with the (weak) "conclusions" text to generate a section "discussion of results and conclusions" or something like that."*

**Author response:**
We have modified the text by moving the intro paragraph to an end summary and conclusion sections. We also feel that we have made a strong conclusion – The VUV lamp can eliminate the use of radioactivity for CIMS.

**24. Referee comment:** *"L255 Extend the conclusions by taking the first paragraph of text from the results section."*

**Author response:**

We have moved the summary paragraph to the conclusion and make it a "summary and conclusion section".

**25. Referee comment:** *"Fig 4 (1) and (3). It would be good to add the ions to the figures. The HNO3I- ion at 1190 amu is the main difference between the Po210 and VUV ion sources and this would be nicely highlighted if the ion-peaks were labelled. In this context, what is the ion at ~ 144 amu (next to I-H2O)? and at ~ 170 amu?"*

**Author response:**

We have added labels to the larger signal ions in the spectra. Please refer to Figure 5 and 8 above in our response to comment 4. There is no peak at 144 but there is a small peak at 143 amu that corresponds to IO$^-$. The peak at ~ 170 corresponds to HCOOH·I$^-$ at m/z 173.

**26. Referee comment:** *"SI: The cross-section of CH3I is x 10-17 (not e17). Give a citation for the cross-section."*

**Author response:**

We have modified the texts as suggested.

**The following are additional minor changes the authors have made to the manuscript:**

1. We corrected the name of one affiliation.

**Page 1 line 7: "²NOAA Chemical Science Laboratory, Boulder, Colorado, USA, 80305"**

**2. We added one new reference for TOF-CIMS**

[revised manuscript text omitted]